# Herpes Simplex Virus Bronchopneumonitis in Critically Ill Patients with Acute on Chronic Liver Failure: A Retrospective Analysis

**DOI:** 10.3390/v16030419

**Published:** 2024-03-08

**Authors:** Miriam Dibos, Julian Triebelhorn, Jochen Schneider, Sebastian Rasch, Roland M. Schmid, Tobias Lahmer, Ulrich Mayr

**Affiliations:** Department of Internal Medicine II, School of Medicine, University Hospital Rechts der Isar, Technical University of Munich, Ismaninger Str. 22, 81675 Munich, Germany

**Keywords:** HSV bronchopneumonitis, HSV reactivation, critically ill, liver cirrhosis, acute on chronic liver failure, intensive care unit

## Abstract

(1) Background: Critically ill patients are frequently diagnosed with pulmonary Herpes simplex virus-1 (HSV) reactivation, which then can lead to HSV bronchopneumonitis and is associated with higher mortality and longer mechanical ventilation. For the particular subgroup of critically ill patients with acute on chronic liver failure (ACLF), however, the impact of HSV reactivation is unknown. We investigated the impact of HSV reactivation in these patients. (2) Methods: We conducted a retrospective analysis, evaluating data from 136 mechanically ventilated patients with ACLF between January 2016 and August 2023. Clinical parameters were compared between patients with and without HSV bronchopneumonitis. (3) Results: 10.3% were diagnosed with HSV bronchopneumonitis (HSV group). Mortality did not differ between the HSV and non-HSV group (85.7% vs. 75.4%, *p* = 0.52). However, the clinical course in the HSV group was more complicated as patients required significantly longer mechanical ventilation (14 vs. 21 days, *p* = 0.04). Furthermore, fungal superinfections were significantly more frequent in the HSV group (28.6% vs. 6.6%, *p* = 0.006). (4) Conclusions: Mortality of critically ill patients with ACLF with HSV bronchopneumonitis was not increased in spite of the cirrhosis-associated immune dysfunction. Their clinical course, however, was more complicated with significantly longer mechanical ventilation.

## 1. Introduction

Severe pneumonia ranks among the primary contributors to mortality in the intensive care unit (ICU), frequently coinciding with the reactivation of Herpes simplex virus-1 (HSV) during the course of treatment [1,2,3]. The frequent incidence of HSV in the upper and lower airway in mechanically ventilated patients may be attributed to the underlying pathogenesis of its reactivation. The reactivation of the latent virus in the oropharyngeal mucosa upon sepsis, stress, and immunosuppression in addition to the trauma of the upper airways during intubation with subsequent micro-aspiration can cause lung parenchyma involvement [4,5]. However, obtaining coherent results in the literature regarding consequences of HSV reactivation and consequent bronchopneumonitis in general populations of critically ill patients proves to be challenging. Mortality of critically ill patients with pulmonal HSV infection has only proved to increase in some studies [3,6,7], while others showed no difference [8,9]. Therefore, it might be necessary to investigate HSV reactivation and subsequent bronchopneumonitis in subgroups of critically ill patients. 

One of these subgroups are patients with acute on chronic liver failure (ACLF). ACLF is a clinical syndrome defined by acute deterioration of a chronic liver disease with the combination of acute decompensation and multiorgan failure [10,11]. Critically ill patients with ACLF are known to have a substantial mortality rate, varying from 32% to 45.3% [10,12]. The increased vulnerability of patients with ACLF has different causes. One reason is cirrhosis-associated immune dysfunction, which results in alterations both in the innate and acquired immunity [13]. Hence, cirrhosis-associated immune dysfunction leads to an increased systemic inflammation, which results in subsequent immunodeficiency. A high predisposition to various infections, especially with Gram-negative bacteria but also to fungal infections, is a well-known precipitator of ACLF and responsible for higher morbidity and mortality rates [11,13,14]. While the roles of bacterial and fungal infections as negative prognostic factors in ACLF patients are generally acknowledged, the possibility of viral infections influencing the clinical course of ACLF lacks evidence. 

Therefore, we conducted a single-center retrospective study investigating the effect of HSV lung reactivation and bronchopneumonitis in critically ill patients with ACLF. The aim was to determine the incidence of HSV bronchopneumonitis in patients with ACLF and to verify its clinical relevance and impact on the outcome of these patients.

## 2. Materials and Methods

### 2.1. Patient Selection

A retrospective study was conducted in a medical ICU at Klinikum Rechts der Isar in Munich, Germany. Between January 2016 and August 2023, all patients with ACLF hospitalized in the ICU requiring mechanical ventilation were eligible for study inclusion. Diagnosis of chronic liver disease was based on sonographic and laboratory evidence. The definition of ACLF was based on the definition from the European Association for the Study of the Liver–Chronic Liver Failure Consortium [15]. All ICU patients requiring mechanical ventilation were screened regularly for bacterial and fungal hospital-acquired pneumonia/ventilator-associated pneumonia (HAP/VAP), and HSV and cytomegalovirus (CMV) reactivation in case of clinical deterioration (increasing inflammatory blood values, febrile temperatures, new/increasing pulmonary infiltrates, or increasing need of oxygen). Only patients with at least one lower respiratory tract sample tested for HSV during their ICU stay were included. Low viral load was defined as a positive HSV polymerase chain reaction (PCR) in the lower respiratory tract with 10^3^–10^5^ IU/mL, whereas high viral load was defined as ≥10^5^ Geq/mL HSV. Clinical deterioration and suspected lung infection in patients with a high viral load of ≥10^5^ Geq/mL HSV in their bronchoalveolar lavage was considered HSV bronchopneumonitis (HSV group). All patients with HSV viral load ≥10^3^ Geq/mL received an antiviral treatment (acyclovir) for 14 days during their ICU course unless they died before initiation or end of treatment (10 mg per kg of body weight with dose adaption according to renal function). Furthermore, bacterial superinfections and also galactomannan antigen in serum and bronchoalveolar lavage (BAL) or non-directed bronchial lavage (NBL) were assessed. For galactomannan antigen and an optical density index (ODI) with a cut-off of >1 for BAL/NBL and >0.5 for serum were suspected for invasive aspergillosis [16]. According to EORTC/MSG-criteria for invasive aspergillosis, additional standard cultures for Aspergillus spp. were obtained [17]. 

### 2.2. Baseline Assessment

Patients’ medical files were screened for age, sex, body mass index (BMI), duration of mechanical ventilation, delay between mechanical ventilation and HSV bronchopneumonitis, other organ failures during ICU stay (especially renal replacement therapy, development of HAP/VAP), ICU and hospital duration and mortality, survival at day 28 and survival at day 90, sepsis-related organ failure assessment score (SOFA score), Child–Pugh score and ACLF score were assessed upon admission and upon development of HSV bronchopneumonitis.

### 2.3. Study Design

BAL or NBL were collected as soon as the patients developed increasing inflammatory blood values, as soon as they became febrile (a temperature of at least 38.5 °C), had new or increasing pulmonary infiltrates on chest roentgenogram or computer tomography scan, or developed an increasing need for oxygen. BAL was conducted by placing the bronchoscope in the wedge position and performing lavage with 10 mL NaCl (0.9%). BAL and NBL fluid samples were processed obtaining bacterial, fungal and viral bronchopneumonitis/reactivation. Bacterial and fungal VAP were defined by bacteriological or fungal evidence of pulmonary parenchymal infection, systemic signs of infection and new or deteriorating infiltrates on chest roentgenogram or computer tomography scan [9].

### 2.4. Statistical Analysis

For data collection, we used the database of our hospital information system (SAP Clinical Data Warehouse Cloud^©^, SAP SE, Walldorf, Germany, Version PKIS 6.17/16). Statistical analysis was performed using Prism 9, (GraphPad PRISM, Boston, MA, USA, Version 9.4.1 (458), 18 July 2022). Samples were tested for normal distribution using the Shapiro–Wilk test. Normally distributed parameters are presented as mean ± standard deviation and, accordingly, descriptive data without normal distribution as median and interquartile range (IQR) and categorial variables as absolute numbers and percentages. For the analysis of quantitative variables, the *t*-test and the Mann–Whitney-U test were employed. All statistical tests were conducted two-sided with a level of significance (*p*-value) of 5%.

## 3. Results

### 3.1. Baseline Characteristics of Patients with ACLF

During the study period, 232 patients with ACLF were treated in our ICU and were eligible for study inclusion (see flowchart, Figure 1). Out of these 232 patients, 155 patients (66.8%) required mechanical ventilation. A total of 19 patients were excluded due to insufficient documentation. All of the remaining 136 patients underwent viral and microbiological testing in respiratory samples (either BAL or NBL) upon clinical deterioration. Of these 136 patients, 26 patients (19.1%) tested positive for at least 10^3^ Geq/mL HSV. Only 14 of these positively tested patients showed high viral load of pulmonary reactivated HSV (≥10^5^ Geq/mL, 10.3%, HSV group).

The reasons for ICU admission for all 136 patients were diverse: 35% of the patients were admitted due to sepsis, 19% due to gastrointestinal bleeding, 18% due to respiratory failure, and 15% due to reduced vigilance (mostly because of hepatic encephalopathy) (see Figure 2).

The main etiology for chronic liver disease was ethyl-toxic (76%), followed by cryptogenic (11%) and viral (9%). Baseline characteristics are summarized in Table 1. 

A total of 14 patients were diagnosed with a high HSV viral load of ≥10^5^ Geq/mL (10.3%, HSV group), while 122 patients did not show a high viral load (89.7%, non-HSV group). Some 12 out of these 122 patients showed a low HSV load (10^3^–10^5^ Geq/mL HSV). To understand the implications of a low HSV load, baseline parameters (median duration of mechanical ventilation, ICU stay, and mortality) between patients without any HSV load and with low viral load were compared. No significant difference was observed between patients with no detectable HSV load and with low viral load (*p* = 0.09 for duration of mechanical ventilation, 0.07 for duration of ICU stay, and 0.3 for mortality). In the following, patients with a low viral load below 10^5^ Geq/mL were therefore included in the non-HSV group. Baseline characteristics such as age, gender, and BMI were similar in the HSV and the non-HSV group (see Table 1). Furthermore, SOFA score, Child–Pugh score and ACLF score at the time of admission were not significantly different between groups.

### 3.2. Characteristics of Patients in the HSV Group

HSV viral load in the HSV group ranged from 142,950 to 40,000,000 Geq/mL, with a median viral load of 2,400,000 Geq/mL. The median time period between intubation and HSV detection was 9.5 days (7.8–17.3). As shown in Table 2, patients in the HSV group had significantly elevated SOFA scores at the time of HSV detection compared to admission (*p* = 0.008). 

### 3.3. Bacterial and Fungal Superinfections in Patients with HSV Bronchopneumonitis

The pulmonal screening at time of respiratory deterioration included screening for viral, bacterial and fungal superinfections. The incidence of bacterial HAP/VAP was not significantly different between the non-HSV and the HSV group (40.5% in the non-HSV group vs. 50% in the HSV group, *p* = 0.57, see Table 1). As the differentiation of the bacterial spectrum of HAP/VAP was not the aim of this paper, it is not differentiated further. 

Fungal superinfections, however, were significantly more frequent in the HSV than in the non-HSV group. Pulmonal mycological cultures for aspergillus spp. were positive in 28.6% (4 out of 14 patients) of the patients with HSV bronchopneumonitis, whereas only 6.6% (8 out of 122 patients) without bronchopneumonitis were screened positive for invasive aspergillosis (*p* = 0.006). Galactomannan antigen in NBL in patients in the HSV group did not differ between groups (4.3 (1.5–5.4) vs. 3.3 (1.5–5.5), *p* = 0.24). Serum galactomannan antigen was only assessed in 22/136 patients and was therefore not included in the statistical analysis.

### 3.4. Clinical Course and Outcome

Patients in the HSV group required significantly longer mechanical ventilation compared to patients in the non-HSV group (21 vs. 14 days, *p* = 0.04, see Table 3). No statistical significance was observed in duration of ICU stay (26 vs. 16.5 days, *p* = 0.11) and in-hospital mortality rates (85.7% vs. 75.4%, *p* = 0.52) between the HSV and Non-HSV group. 

### 3.5. Survival Time of Patients in the HSV and Non-HSV Group

Among the 136 patients with ACLF, HSV bronchopneumonitis did not significantly impact ICU survival (Figure 3). 

### 3.6. Impact of Cytomegalovirus Reactivation on Survival

All of the patients in the HSV group were also tested for CMV reactivation. CMV was not detected in any of these patients (see Table 1). 

## 4. Discussion

With an incidence of 10.3% of HSV bronchopneumonitis in our cohort of patients with ACLF, we observed a relevant but comparatively lower incidence rate in this special subgroup. Reported rates of HSV bronchopneumonitis in critically ill patients in the literature vary between 22 and 62% [4,18,19]. 

One reason for the frequent diagnosis of HSV bronchopneumonitis in ICU patients in general is the lack of a precise diagnostic standard for HSV bronchopneumonitis and the ill-defined cut-off for the viral load of HSV at which antiviral treatment should be initiated [20,21]. The lack of a diagnostic standard due to the lack of randomized controlled studies results in potential over-diagnosis and also over-treatment [21]. According to multiple retrospective studies, only patients with high viral load (≥10^5^ Geq/mL) show increased mortality and may benefit from treatment with acyclovir [7,18]. Rather than having any significant influence on prognosis, low quantities of HSV (<10^5^ Geq/mL) are more likely to be the result of a reactivation brought on by immunosuppression, stress, sepsis, and trauma [18,22]. In our study group, we did not find any significant differences in duration of mechanical ventilation, ICU stay, or mortality when compared with the low-viral-load group (<10^5^ Geq/mL) to the patients without pulmonary HSV reactivation. This aligns with research indicating that antiviral therapy does not impact the total duration of stay in the ICU or hospital, as well as mortality, among patients with low viral load [18]. Nonetheless, due to the fact that patients with ACLF are an extremely vulnerable subgroup of ICU patients due to cirrhosis-associated immune dysfunction, all but one of the patients in the low-viral-load group received treatment with acyclovir.

The objectives of our study were to investigate predictive factors as well as the impact of pulmonary HSV reactivation in a vulnerable subgroup of mechanically ventilated patients with ACLF. To determine potential predictive factors for the development of HSV bronchopneumonitis, we compared infection parameters (C-reactive protein, procalcitonin, white blood cells) at time of admission vs. at time of HSV detection and found no statistically significant differences (see Table 1). However, the median SOFA score in patients in the HSV group was significantly higher at the time of HSV diagnosis than at admission, at 17.5 vs. 14 (*p* = 0.008). Furthermore, patients in the HSV group required longer mechanical ventilation (21 vs. 14 days, *p* = 0.04), while the total duration of ICU treatment and mortality did not differ between the HSV and non-HSV group. Nonetheless, prolonged mechanical ventilation alone leads to higher economic costs and therefore a higher financial burden on the health care system. The prolonged mechanical ventilation of patients with HSV bronchopneumonitis is consistent with other studies of mechanically ventilated patients with HSV bronchopneumonitis, which shows an association between HSV reactivation and prolonged mechanical ventilation [1,8]. However, it is still debated within the scientific community whether HSV reactivation itself is associated with increased morbidity and mortality or whether it should be considered a bystander association (as it is usually diagnosed in the most severe patients) [1]. As patients in our study group developed HSV bronchopneumonitis after a median ventilation duration of 9.5 days [7.8–17.3], patients with prolonged mechanical ventilation and prolonged weaning in particular require more frequent screening for pulmonal HSV and fungal infections upon clinical deterioration. Therefore, we recommend bacterial, viral, and fungal respiratory screening in all critically ill patients upon clinical deterioration. Clinical deterioration includes increasing inflammatory blood values, febrile temperatures, new/increasing pulmonary infiltrates, and increasing need for oxygen. Patients with prolonged weaning and/or immunocompromised patients (including patients with chronic liver disease, hemato-oncological diseases, patients on immunosuppressive drugs, etc.) in particular should undergo regular screening, as already suggested by our group [23]. In case of septic multiorgan failure and HSV bronchopneumonitis with an HSV viral load of ≥10^5^ Geq/mL, we recommend treatment with acyclovir and continuation of therapy despite the general high mortality rate.

A typical reason for prolonged mechanical ventilation in ICU patients is in most cases HAP/VAP caused by bacterial super and/or co-infection. In our study, bacterial superinfections could be detected in 41.1% of patients without a significant difference between the non-HSV and HSV group (40.5% in non-HSV group vs. 50% in HSV group, *p* = 0.57, see Table 1). As the bacterial spectrum of HAP/VAP was not a goal of our study, it is not differentiated further in this paper. Alongside the most common bacterial super/co-infections in critically ill patients, awareness of invasive fungal infections, especially the development of invasive aspergillosis beyond the typical risk groups, has been increasing in recent years [24]. Although little is known about invasive aspergillosis in critically ill patients with ACLF, Lahmer et al. showed that the risk of invasive aspergillosis in these patients is significantly increased and associated with a prolonged length of ICU stay and higher mortality and therefore has a significant impact on outcomes [24]. This and recently published papers concerning invasive aspergillosis as a super/co-infection in other viral pneumonias (especially in COVID-19 and influenza pneumonia) compelled us to screen our high-viral-load HSV cohort for invasive fungal infections [25,26]. The incidence of probable invasive aspergillosis including positive mycological cultures and elevated BAL galactomannan was significantly higher in the HSV group than in the non-HSV group (28.6% vs. 6.6%, *p* = 0.006, see Table 1). Nonetheless, absolute ICU mortality did not differ between the HSV group (12 out of 14 patients, 85.7%) and the non-HSV group (92 out of 122 patients, 75.4%, *p* = 0.52, see Table 3). 

We expected HSV bronchopneumonitis to be more common in critically ill patients with ACLF than in other populations of ICU patients because of cirrhosis-associated immune dysfunction. The relatively low incidence may be attributed to the scarcity of studies on the occurrence of HSV bronchopneumonitis within subgroups of ICU patients (e.g., ICU patients with hematooncological diseases, on immunosuppressive drugs, after organ transplantation), as most studies have investigated the effect in general populations of critically ill patients or ICU patients with ARDS [3,4,18,19]. Another reason may be that our population of patients with ACLF included a variety of different acute diseases ranging from acute variceal hemorrhage to hepatic encephalopathy to sepsis. Patients with ACLF who are admitted due to acute variceal hemorrhage show a different degree of immunosuppression compared to patients with ACLF admitted due to sepsis. We expect the latter to have a significantly increased risk of viral reactivation due to the additional immunosuppression but decided against another subgroup analysis due to the small size of the group. 

Our findings that ICU mortality was not increased in the HSV group could be explained by a variety of factors. One factor is the fact that our study only included the most severely ill patients with ACLF, multiple organ dysfunctions or failures, and generally very high mortality. Overall mortality of critically ill patients with ACLF in the literature varies from 32% to 45.3% [10,12], while mortality in our cohort including all 232 patients with and without mechanical ventilation was 56.9%. As we only included patients with respiratory failure and mechanical ventilation, mortality was even higher (76%, 104 out of 136 patients). Another factor to be considered may be that patients were included due to their chronic liver disease and not due to ARDS criteria. Pneumonia was the main reason for ICU admission in only 18% of our patients. This may explain the relatively rare incidence of HSV bronchopneumonitis. Studies which included only critically ill patients with severe ARDS, e.g., due to COVID-19, or influenza patients with severe ARDS found significantly higher rates of HSV bronchopneumonitis [4,18,19]. Studies which included critically ill patients without screening for ARDS found a lower prevalence of HSV bronchopneumonitis compared to the studies mentioned above, but still found higher rates than in our cohort. A study by Ong et al. detected HSV viral load in 106 out of 393 patients (27%) [27]. Another study by Luyt et al. found HSV bronchopneumonitis in 21% of all critically ill patients who deteriorated clinically (including medical and surgical postoperative ICU patients) [9].

The following limitations have to be addressed. First, due to the nature of our retrospective analysis, amongst others, HSV serology, galactomannan antigen in NBL, BAL and serum were not assessed for all patients included in the study. Second, as mentioned above, patients were not admitted primarily due to ARDS, which may explain the relatively low incidence of HSV bronchopneumonitis. Third, we included an extremely vulnerable group of critically ill ACLF patients with high mortality rates. In this cohort of critically ill patients, HSV bronchopneumonitis did not aggravate already very high mortality rates. This could be explained by the fact that HSV bronchopneumonitis is an expression of illness severity rather than having its own prognostic worth. Fourth, we only included patients on mechanical ventilation as we generally only perform BAL with microbiological and virological testing in patients with suspected VAP. Fifth, we only included patients who underwent testing for HSV, which excluded 19/155 patients (12%) due to insufficient documentation.

## 5. Conclusions

In summary, in our retrospective analysis, HSV bronchopneumonitis appears to be a rather infrequent complication in patients with ACLF, in contrast to other ICU populations. Although patients with ACLF and HSV bronchopneumonitis required longer mechanical ventilation, the diagnosis did not affect the outcome of these patients. Mortality in our cohort was extremely high as only the most severely ill patients were included. In this cohort of critically ill ACLF patients, HSV bronchopneumonitis might only be a marker of illness severity and should probably be treated. We therefore propose screening patients with chronic liver disease and ACLF who require long periods of mechanical ventilation or protracted weaning and suffer clinical deterioration. Diagnostic testing should include screening for pulmonary bacterial, viral, and fungal infections/co-infections, even if patients did not present with pneumonia primarily. In cases of septic multiorgan failure with rising SOFA score and an HSV viral load of ≥10^5^Geq/mL, we recommend treatment with acyclovir and continuation of therapy despite the generally high mortality rate.

## Figures and Tables

**Figure 1 viruses-16-00419-f001:**
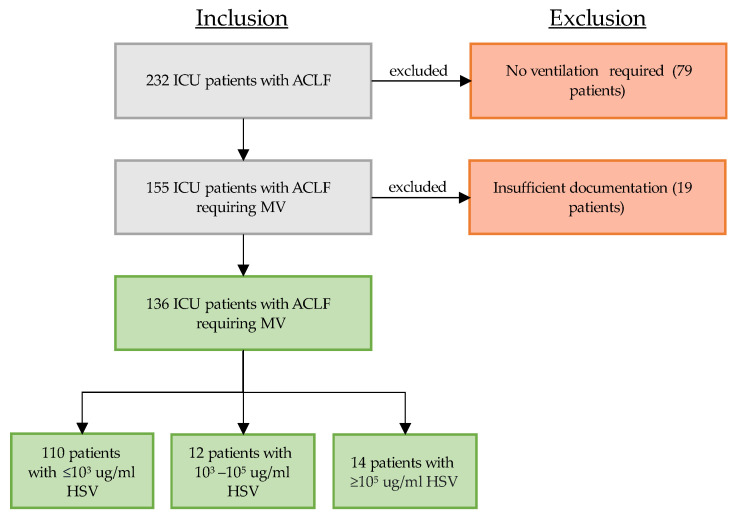
Patient selection between 2016 and 2023: critically ill patients with ACLF requiring mechanical ventilation who received quantitative real-time polymerase chain reaction (PCR) testing of their respiratory material (bronchoalveolar lavage or non-directed bronchial lavage). Grey box indicates ongoing selection process, green box indicates inclusion, red box indicates exclusion, arrows to the right indicate exclusion. Abbreviations: ICU: intensive care unit, ACLF: acute on chronic liver failure, MV: mechanical ventilation.

**Figure 2 viruses-16-00419-f002:**
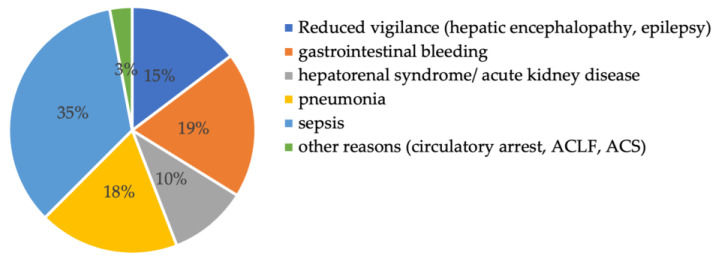
Reason for intensive care unit admission. Abbreviations: ACLF: acute on chronic liver failure, ACS: acute coronary syndrome.

**Figure 3 viruses-16-00419-f003:**
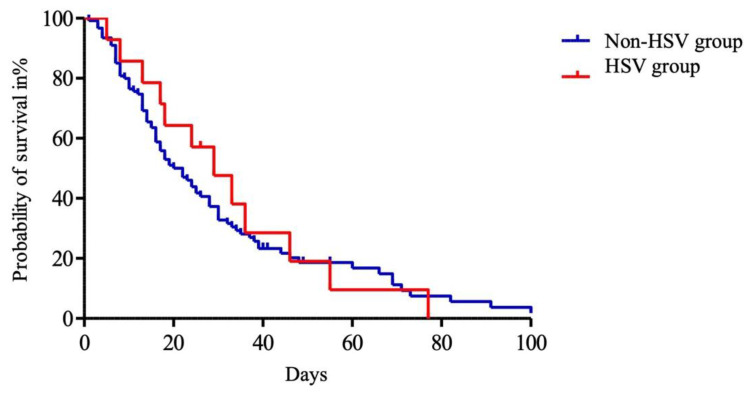
Kaplan–Meier analysis of intensive care unit (ICU) survival of patients with acute on chronic liver failure. Day 0 was defined as ICU admission. The entire cohort was divided into HSV group (red) and non-HSV group (blue). Abbreviations: HSV: herpes simplex virus.

**Table 1 viruses-16-00419-t001:** Baseline characteristics of patients in the HSV and non-HSV group. Results are presented as mean (SEM), median [IQR] or *n* (%).

Parameters	Total (136 Patients)	Non-HSV Group (122 Patients)	HSV Group (14 Patients)	*p*-Value
Upon admission:				
Median age—years (IQR) Male sex—% (*n*) Median BMI (IQR) Median SOFA score (IQR) Median Child—Pugh score (IQR) Median ACLF score (IQR)	57 (51–67) 64% (87) 25 (22–28) 13 (9–15.8) 12 (10–13) 3 (2–3)	57 (50–67) 61.5% (75) 25 (22–28) 13 (9.8–15.3) 12 (10–13) 3 (2–3)	63 (55–69) 85.7% (12) 24.5 (22.8–27.3) 14 (8.8–16.3) 11.5 (9–14) 3 (1.8–3)	0.28 0.09 0.95 0.97 0.96 1
Reason for mechanical ventilation				
- Respiratory failure - Neurologic failure - Cardiac arrest - Gastrointestinal bleeding - Sepsis - Post-operative	41.2% (56) 19.9% (27) 6.6% (9) 21.3% (29) 9.6% (13) 1.5% (2)	41.8% (51) 20.5% (25) 7.4% (9) 19.7% (24) 9% (11) 1.6% (2)	35.7% (5) 0% (0) 14.3% (2) 35.7% (5) 14.3% (2) 0% (0)	
Reason for chronic liver disease				
- Ethyl-toxic - Viral - Autoimmune - NASH - Cryptogenic	76.5% (104) 8.8% (12) 1.5% (2) 2.2% (3) 11% (15)	73.8% (90) 9.8% (12) 1.6% (2) 2.5% (3) 12.3% (15)	100% (14)	
Bacterial HAP/VAP–% (*n*)	41.1% (56)	40.5% (49)	50% (7)	0.57
Probable invasive aspergillosis (GM-antigen ≥1) Mean galactomannan (ODI) (*n*) Cultural microbiological evidence	21.8% (22/101) * 3.8 (1.6–5.5) 11.9% (12/101)	19.3% (17/88) * 3.3 (1.5–5.5) 6.6% (8/122)	38.5% (5/13) * 4.3 (1.5–5.4) 28.6% (4/14)	0.1 0.24 0.006 (*)
Dialysis—% (*n*)	78.7% (107)	77% (94)	92.9% (13)	0.3
Infection parameters at admission:				
- C-reactive protein in mg/dL - Procalcitonin in ng/mL Leukocytes in G/L	5 (2.8–7.2) 1.3 (0.5–3.2) 12.3 (8.6–17.4)	5 (2.8–7.2) 1.3 (0.4–3.1) 12.3 (8.5–17.5)	4.5 (2.9–8.2) 1.1 (0.5–5.1) 12.7 (7.7–23.5)	0.94 0.83 0.68
Pulmonal CMV ≥10^5^ Geq/mL—% (*n*)	3% (4/133)	3.3% (4/122)	0% (0/14)	

Abbreviations: SD: standard deviation, BMI: body mass index, IQR: interquartile range, SOFA: sepsis-related organ failure assessment score, ACLF: acute on chronic liver failure, NASH: non-alcoholic steatohepatitis, HAP: hospital-acquired pneumonia, VAP: ventilator-associated pneumonia, GM: galactomannan, * Galactomannan antigen was only obtained for 101 patients in total, with (x/x) showing the *n*-count for which GM antigen was obtained and (*) indicating statistical significance.

**Table 2 viruses-16-00419-t002:** Baseline characteristics of patients in the HSV group at admission vs. at the time of HSV detection.

Parameters	At Admission	At HSV Detection	*p*-Value
Median SOFA score (IQR) Median Child—Pugh score (IQR) Median ACLF score (IQR)	14 (8.8–16.3) 11.5 (9–14) 3 (1.8–3)	17.5 (14.8–20.3) 12 (10.8–13) 3 (3–3)	0.008 (*) 0.95 0.16
Infection parameters:			
- C-reactive protein in mg/dL - Procalcitonin in ng/mL - White blood cells in G/L	4.5 (2.9–8.2) 1.1 (0.5–5.1) 12.7 (7.7–23.5)	5.2 (3.1–6.7) 2.1 (0.7–11.6) 14.2 (8–20.8)	0.79 0.36 0.98

Abbreviations: SOFA: sepsis-related organ failure assessment score, IQR: interquartile range, ACLF: acute on chronic liver failure, (*) indicating statistical significance.

**Table 3 viruses-16-00419-t003:** Clinical course of patients in the HSV and non-HSV group.

Parameters	Total	Non-HSV Group	HSV Group	*p*-Value
Total duration of MV, d (IQR)	14.5 (8–27.3)	14 (7.3–26.8)	21 (11.8–39.3)	0.04 (*)
Total duration of ICU-treatment, d (IQR)	17.5 (10–31.5)	16.5 (10–30)	26 (16–38.5)	0.11
In-hospital mortality %, (n) - **Death day 28** - **Death day 90**	76.5% (104) 53.7% (73) 75% (102)	75.4% (92) 54.9% (67) 73.8% (90)	85.7% (12) 42.9% (6) 85.7% (12)	0.52 0.41 0.52

Abbreviations: MV: mechanical ventilation, IQR: interquartile range, ICU: Intensive care unit, (*) indicating statistical significance.

## Data Availability

No new data were created or analyzed in this study. Data are contained within the article.

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
