# Peer review of "Herpes Simplex Virus Bronchopneumonitis in Critically Ill Patients with Acute on Chronic Liver Failure: A Retrospective Analysis"

_viruses, 2024, doi:10.3390/v16030419_

Round 1

Reviewer 1 Report

Comments and Suggestions for Authors

The manuscript reports on a retrospective analysis of HSV bronchopneumonitis in patients with acute on chronic liver failure (ACLF).  The study determined that HSV bronchopneumonitis was not increased in ACLF patients despite a state of immunosuppression associated with cirrhosis-associated immune dysfunction.  The justification for the retrospective study is logical and the outcome, even if not finding a stronger association of HSV bronchopneumonitis in the ACLF patients, is a valuable finding.  The inclusion of the parameters outlined by the authors in the discussion would provide for a more in-depth analysis of the association between HSV bronchopneumonitis and the impact on hospitalization in this group.

The methodology is appropriate and the statistical analysis is valid.

Comments on the Quality of English Language

The quality of English Language is excellent, with some minor edits required.

Author Response

Dear reviewer,

Best regards,
Miriam Dibos

Reviewer 2 Report

Comments and Suggestions for Authors

This is an interesting and well written manuscript examining the incidence of herpes simplex virus (HSV) respiratory infections in critically ill acute on chronic liver failure patients  with acute respiratory failure.  Those with high levels of HSV in respiratory samples had a longer duration of mechanical ventilation and longer ICU stay, likely due to the longer period of intubation.  There were no differences in mortality in the groups, which is not surprising as ICU mortality is related to several factors.  The authors make a point that only high levels of HSV infection should be treated as lower levels of infection did not affect outcomes.

A surprising finding from the manuscript is that the incidence of HSV infection in critically ill acute on chronic liver failure patients is much lower than that seen in other critical care studies.  One would expect a higher incidence in this significantly immunocompromised population.  This significant difference is only partially discussed in the manuscript.  This needs to be expanded upon.  Additional discussion regarding the role of HSV screening in all critically ill patients with respiratory failure would add to the manuscript.  Questions on who should be screened, when screening should occur, how frequent patients should be screened, and how to interpret the results as it relates to treatment would be helpful to the critical care community as a whole.

Author Response

(The authors gave the same response as above.)

Reviewer 3 Report

Comments and Suggestions for Authors

Dear authors , I was happy to review the manuscript with the title "Herpes simplex virus bronchopneumonitis in critically ill patients with acute on chronic liver failure: a retrospective analys".

The manuscript describes critically ill patients  with  Herpes simplex virus-1 (HSV) reactivation. The diseases presented are HSV bronchopneumonitis (associated with higher mortality and longer mechanical ventilation) and acute on chronic liver failure. The study describes 136 mechanically ventilated patients from  January 2016 to August 2023, comparing parameters  between patients with and without HSV bronchopneumonitis.  The methods and the study design is fine.

Introduction section and the cited references must be improved. More studies must be discussed in the disscution section too.

Results may be improved also, I believe figure 1 may be presented in a different manner.

Line 149 has a strange text, please rephrase, also line 166.

 The conclusions support the results.

(3) Results: 10.3% were diagnosed with HSV broncho- 19 pneumonitis (HSV group). Mortality did not differ between the HSV and non-HSV group (85.7% 20 vs. 75.4%, p = 0.52). However, the clinical course in the HSV group was more complicated as patients 21 required significantly longer mechanical ventilation (14 vs. 21 days, p = 0.04). Furthermore, fungal 22 superinfections were significantly more frequent in the HSV group (28.6% vs. 6.6%, p = 0.006). (4) 23 Conclusions: Mortality of critically ill patients with ACLF with HSV bronchopneumonitis was not 24 increased in spite of the cirrhosis-associated immune dysfunction. Their clinical course however 25 was more complicated with significantly longer mechanical ventilation

Author Response

(The authors gave the same response as above.)

Round 2

Reviewer 2 Report

Comments and Suggestions for Authors

No additional comments

Reviewer 3 Report

Comments and Suggestions for Authors

I am ok with the reviewed version.